# Quality Ingredients and Safety Concerns for Traditional Fermented Foods and Beverages from Asia: A Review

**Anil Kumar Anal** 

Food Engineering and Bioprocess Technology, Department of Food, Agriculture and Bioresources, Asian Institute of Technology, PO Box 4, Klong Luang, Pathumthani 12120, Thailand; anilkumar@ait.ac.th

**Abstract:** Fermented foods and beverages serve as vehicles for beneficial microorganisms that play an important role in human health and remain the oldest prevalent means of food processing and preservation. Traditional fermented foods are popular in Asia for their nutritional balance and food security. Techniques for preserving cereals, vegetables, and meat products are well developed in many Asian countries. Due to their cultural and nutritional significance, many of these foods have been studied in detail and their quality and safety have also been improved. These fermented foods and beverages provide benefits through enhanced nutritional content, digestibility, microbial stability, and detoxification. They represent is thus one of the most affordable and suitable methods to maintain hygiene condition and food quality and security in poor and underdeveloped countries. There is an industrial interest and scope related to traditional fermented foods and beverages in Asia. However, urgent attention is required to improve the quality of the ingredients and the integration of food safety management systems for industrial growth.

**Keywords:** fermentation; traditional; nutritional value; microbiology; Asian countries

## 1. Introduction

Indigenous fermented foods and beverages have been part of the human diet since the beginning of civilization. Such foods are either served as staples or adjuncts to staples, pickles, condiments, and beverages. Fermentation dates back to the Neolithic period (circa 10,000 B.C.), when it was used primarily for the preservation of perishable food. Microorganisms were generally inoculated to use for the fermentation and maturation of foods. Fermentation was primarily aimed for food preservation, obtained by the formation of inhibitory metabolites, such as organic acid, ethanol, and bacteriocins, in combination with reduced water activity. However, fermentation has been explored for other functions such as improvement in food safety through the inhibition of pathogens or removal of toxic compounds; improvement in nutritional value; and improvement in the organoleptic quality of the food [1]. In addition, fermentation provides a natural way to reduce the volume of the material to be transported, to destroy undesirable components, to enhance the nutritive value and appearance of the food, to reduce the energy required for cooking, and to make a safer product.

Fermented food products are produced widely using different techniques, raw materials, and microorganisms. However, there are basically only four types of fermentation processes involved in the product development, namely, alcoholic, lactic acid, acetic acid, and alkali fermentation, as described below.

Lactic acid fermentation is mainly carried out by lactic acid bacteria (LAB). Examples include fermented cereals, *kimchi*, sauerkraut, and *gundruk*.

Alcohol fermentation contributes to the production of ethanol. Yeasts are the predominant organisms, for example, wines, beers, vodka, whiskey, brandy, and bread.

Acetic acid fermentation is produced from the *Acetobacter* species. *Acetobacter* converts alcohol to acetic acid in the presence of oxygen (e.g., vinegar).

Alkaline fermentation takes place during the fermentation of soybeans, fish, and seeds, popularly used as a condiment.

Fermented foods are associated with a unique group of microflora that enhance the nutritional quality of food such as proteins, vitamins, essential amino acids, and fatty acids. On the other hand, three quarters of humanity are deprived of basic food and are malnourished. In this regard, fermented food products can address the problems related to the world's balanced diet [2]. Fermented food products are typically unique and vary depending on the region due to the variation in environmental conditions, cultural and social aspects, taste preferences, availability of raw materials, and new technological development [3]. Based on raw materials available, different types of fermented food products are prepared to increase food varieties in order to overcome food and nutrition insecurity.

Asia is well known for its techniques to preserve and balance the fluctuation in food availability during the monsoonal circulation. Paddy production in Southeast Asian countries accounts for 25% of world production (150 million tons per year) of which 95% is consumed within the region. The fermentation of cereals and other plant products to produce a variety of foods is a common practice since ancient times. Rice wine is one of the popular fermented beverages and fermented cassava tubers is another of the many fermented food products widely consumed in Asian countries [4].

Fermented foods and beverages may vary based on the nature of the food, the fermentation time, and the intentional application of microbes utilized. Fermentation could occur at the starting process for products that undergo multiple additional steps once the fermentation is terminated, such as in coffee, chocolate, tea leaves, sourdough bread products, among others (Table 1). On the other hand, fermentation could continue to characterize the final products, such as in pungent-smelling blue cheese or vinegar-tasting kombucha [5].

**Table 1.** Examples of common fermented foods and beverages developed during process and/or as final products.

| Fermentation in the Preparation Phase | Fermentation of the Final Food Product |
|:---:|:---:|
| Coffee | Soy Sauce |
| Chocolate | Yogurt |
| Fermented tea leaves | Kimchi |
| Sourdough bread | Kombucha, beer, wine |

Fermentation is utilized in the preparation phase typically in products such as chocolate and coffee. Chocolate is made from the fermentation of cocoa beans with the successive action of yeast, acetic acid bacteria (AAB), and lactic acid bacteria (LAB) driving the conversion of pulp substrate into ethanol, lactic acid, and acetic acid. During fermentation, flavor and aroma precursors develop and pigments are degraded by the action of enzymes such as invertases, glycosidases, proteases, and polyphenol oxidase. Fermentation influences the compounds such as reducing sugars, peptides, and amino acids, which are converted into flavor and smell profiles during the drying and roasting steps of chocolate processing [6]. Tea is a rich source of several flavonoid compounds that are responsible for its distinctive taste and color along with various health benefits. In a research study by Jayasekera et al. [7], total catechins, total flavonols, and total theaflavins were observed to be higher in fermented Sri Lankan tea leaves compared to the unfermented tea leaves. Traditionally, coffee beans are fermented to remove mucilage and prepare the beans for roasting. However, the fermentation also helps in the development of the coffee's aroma quality.

Figure 1 illustrates the schematic representation of types of fermentation, the microorganisms involved, and the resulting end products. Based on raw materials, manufacturing techniques, and microorganisms, various types of fermented food products are available. During alcoholic fermentation, yeasts are the predominant microorganisms and result in the production of ethanol.

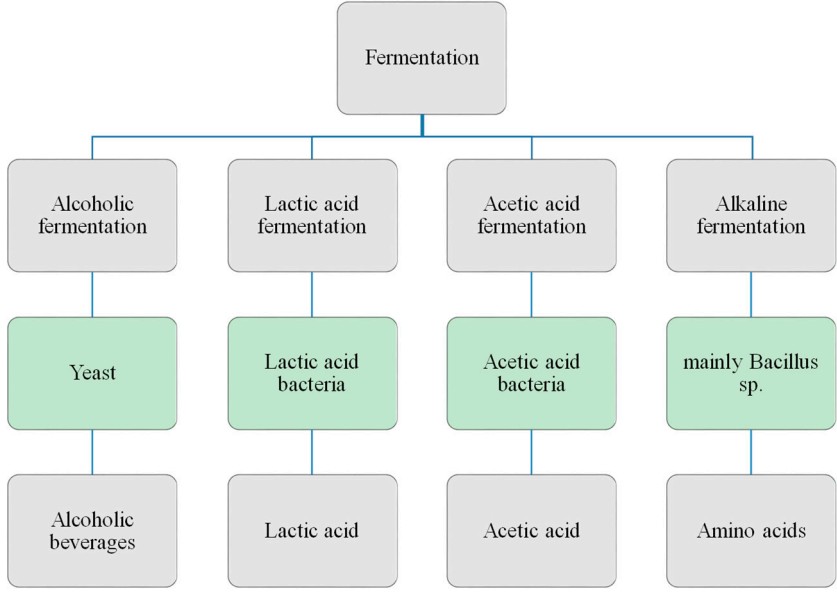

**Figure 1.** Schematic representation of the common types of fermentation, the microorganisms involved, and the end products.

Lactic acid fermentation is mainly carried out by lactic acid bacteria (LAB). The acetic acid fermentation by acetic acid producers from the *Acetobacter* species converts alcohol to acetic acid in the presence of excess oxygen. Alkali fermentation often takes place during the fermentation of fish and seeds, popularly used as condiments [8].

## 2. Quality Attributes of Common Fermented Foods of Asia

The prevalence of malnutrition and nutritional deficiencies are common problems in developing and underdeveloped countries. The lack of proper dietary quality, an inadequate food consumption, a lower nutrient bioavailability, and the outbreak of infectious diseases are considered to be the major reasons for malnutrition. Furthermore, concerns about food safety regarding microbial contamination and naturally occurring toxins are of great importance. To combat these problems, simple, cost-effective, and indigenous food-based approaches such as the fermentation process are considered to be effective. The fermentation process improves the nutritional quality, digestibility, and bioavailability of nutrients, while also reducing anti-nutritional factors and enhancing the shelf life and safety of the product [9].

### 2.1. Alcoholic Fermented Foods and Beverages

Alcoholic fermentation has been practiced since ancient times and is therefore one of the oldest and most important techniques in food processing. This process results in the production of various alcoholic beverages like beers, wines, and distilled liquors, using yeast or sometimes yeast-like molds, such as *Amylomyces rouxii*, and mold-like yeasts such as *Endomycopsis* and bacteria such as *Zymomonas mobilis*. These microorganisms involve the utilization of fermentable sugars from substrates such as cereal grains, sugar cane juice, palm sap, fruit juices, diluted honey, or hydrolyzed starch, resulting in the production of mainly ethanol and carbon dioxide.

In Asia, there are at least two additional ways of fermenting starch rice to obtain alcohol: first, the starch is converted to simple sugars by the action of amylase enzymes produced by mold such as *Amylomyces rouxii*, followed by the fermentation of sugars to ethanol by the action of yeast such as *Endomycopsis fibuliger*. Rice, inflorescences of palm such as coconut and *talipot* palm, millet, and others are common substrates for the production of alcoholic beverages.

The wines from coconut and *talipot* palm are commonly produced all over Asia. The fermented beverage from palm saps is called "*panam culloo*" in Vietnam, "*arak*" in Indonesia, and "*tuak*" or toddy

in Malaysia, India, and Bangladesh. Toddy is the traditional fermented alcoholic beverage made by fermenting the sap from the coconut, palmyra, and toddy palm. The tip of an unopened flower is sliced, causing sap to ooze out from the cut and which is then collected in an earthenware container tied underneath the flower. The sap can be consumed either as fresh sap or allowed to ferment for up to 24 h. The fresh sap is a dirty-brown sweet liquid with a sugar content of approximately 10%–18% *w/w* which on formation results in an alcoholic beverage with an ethanol content of as much as 9% (*v/v*) [10]. Palm wine is one of the cheapest sources of vitamin B for the poorer communities; it contains 0.019%–0.028% (*w/v*) of vitamin B-12 and 0.008% of ascorbic acid. The amounts of thiamine, riboflavin, and pyridoxine increase with fermentation [4]. Fermented coconut toddy contains approximately 1.8–7.9 g alcohol, 0.29 g sucrose, 0.9–3.0 g invert sugar, and 3.72 g/dL total solids. It contains nitrogen, phosphorus, potassium, calcium, and magnesium <0.5 g/100 mL [11]. The palm sap fermentation process involves alcoholic-lactic-acetic acid fermentation in the presence of mainly yeasts and LAB. The yeast species *Saccharomyces cerevisiae* is invariably present along with other LAB such as *Lactobacillus plantarum*, *Lactobacillus mesenteroides*, or other species of bacteria such as *Zymomonas mobilis* and *Acetobacter* spp. [10]. In coconut sap fermentation, *Saccharomyces chevalieri* is the main alcohol producer [11]. The common traditionally fermented alcoholic beverages in Asia are summarized in Table 2.

**Table 2.** Common fermented alcoholic beverages in Asia.

| Country | Raw Materials | Local Name | Microorganism | References |
|---------|---------------|------------|---------------|------------|
| Bangladesh | Rice<br>Palm | *Bangla maad*<br>*Tari* | Yeast<br>*S. cerevisiae*; *L. plantarum*,<br>*L. mesenteroides*;<br>*Zymomonas mobilis*,<br>*Acetobacter* spp. | [10] |
| Sri Lanka | Coconut, palmyra,<br>or *Caryota urens*<br>flower sap | Toddy | *S. exiguus*, *S. cerevisiae*,<br>*Pichia fermentans*, *S. rosei*,<br>*S. fructuum*, *Torulopsis holmii*,<br>*Torulopsis versatilis*, *Candida robusta*,<br>*Candida lambica*, *Saccharomyces*,<br>*Schizosaccharomyces*, *Brettanomyces* | [11] |
| Nepal | Millet, rice,<br>buckwheat | *Jaand* | *Aspergillus oryzae*, LAB,<br>ethanol-fermenting yeast | [12] |
| | Millet, rice,<br>buckwheat | *Rakshi* | Saccharifying molds, LAB,<br>ethanol fermenting yeast | |
| | Rice | *Hyaun thon* | *Aspergillus oryzae*, *Rhizopus* spp.,<br>LAB | |
| Thailand | Glutinous rice,<br>Loogpaeng | *Sato* (rice wine) | *Amylomyces* sp., *Aspergillus* sp.,<br>*Rhizopus* sp., *S. cerevisiae* | [13] |
| Japan | Rice | *Sake* | *A. oryzae*, *S. cerevisiae*, *L. plantarum*,<br>*L. leichmanii* | [14] |
| Vietnam | Rice<br>Glutinous rice | *Ruou de*<br>*Ruou nep* | *Mucor* spp., *Rhizopus* spp.,<br>*Aspergillus* spp., *S. ellipsoideus*,<br>*S. cerevisiae*, *Endomycopsis fibuliger*,<br>*Hansenula anomala*,<br>*Torulopsis candida* | [15] |

Some of the traditional alcoholic beverages of Nepal include *Jaand*, *Rakshi*, and *Huaun thon*. *Jaand* or *jaanr* is an undistilled alcoholic beverage made from different raw materials such as rice, millet, maize, wheat, and other starch-rich substrates by the action of a starter known as *murcha*. The distilled alcoholic beverage with a characteristic aroma from *jaand* is known as *rakshi*. The major microorganisms present in *murcha* are filamentous molds such as *Mucor circinelloides*, *M. hiemalis*, *Rhizopus chinensis*, and *R. stolonifer* var. *lyococcus*; yeasts such as *S. fibuligera*, *S. capsularis*, *Pichia anomala*, *P. burtonii*, *S. cerevisiae*,

*S. bayanus*, and *C. glabrata*; and LAB such as *P. pentosaceus*, *Lactobacillus bifermentans*, and *L. brevi*. *Jaand* contains 5%–9% alcohol, 0.8%–1.1% acidity (as lactic acid), 1.6%–2.5% reducing sugar as glucose, 1.6%–2.8% total sugar (as sucrose), 12%–14% starch, and 76%–80% water. In the final fermented product, protein content increased up to 17.6%–38.8%; carbohydrate content decreased 86.41% to about 77.29%–77.71%; thiamine content increased up to 16%–32%; pyridoxine content increased by 50%–59%; niacin content increased up to 117%–173%; and the increase in folic acid was nearly 76% [12].

In Thailand, rice is the main crop and is harvested one or two times per year. *Sato* is the traditional alcoholic beverage of Thailand prepared from glutinous rice, a starter culture (*Loogpaeng*), and water. *Loogpaeng* as the starter culture contains different kinds of microorganisms such as fungi, yeast, and bacteria. Dominant fungi are *Aspergillus*, *Mucor*, *Rhizopus*, *Amylomyces*, and *Penicillium*, while *Endomycopsis* and *Saccharomyces* are the existing yeasts in *Loogpaeng*. Along with these, acetic acid bacteria (AAB), LAB and *Bacillus* are also detected in the starter culture. During *sato* production, once the cooked glutinous rice is mixed with *Loogpaeng*, it starts to become turbid and sweet due to the saccharification caused by the amylase enzyme produced by *Amylomyces* and *Rhizopus*. After the addition of water to the sweet rice, yeast including bacteria starts to grow leading to alcoholic fermentation. Depending on the quality of *Loogpaeng*, alcohol content is about 7%–10% (*v/v*) in the final fermented product [13]. *Khoaw Maak* is another popular traditionally fermented Thai dessert made from fermented rice. It is acidic sweet in taste and white in appearance. A research study was conducted to evaluate the bioactive components of the Thai rice fermentation. *Oryza sativa* L. var. *indica*, including white plain, purple plain, brown plain, white glutinous, and purple glutinous rice, was fermented with Look-Pang (a mixed culture of yeasts and molds). The sap sample from the fermentation of the purple plain rice showed the highest free radical scavenging (the sample concentrations that scavenged 50% of the DPPH radicals, $SC_{50}$ at 14.51 ± 2.21 mg/mL), as well as tyrosinase inhibition (the sample concentrations that inhibited 50% of tyrosinase activity, $IC_{50}$ at 15.05 ± 2.92 mg/mL) and MMP-2 inhibition activities (62.22% ± 3.78%). Tyrosinase is the main enzyme that catalyzes the melanin synthesis. Matrix metalloproteinases (MMPs) are enzymes which degrade the collagen matrix, in addition to reducing arthritis, inflammation, heart-related diseases, cancer, and skin aging. Many MMP inhibitors are basically vitamin C and vitamin E [16].

Similar traditional rice wines have equivalent appellations, such as Japanese *Sake* and *Mirin*, Vietnamese *Ruou*, Philippine *Tapuy*, Indonesian *Brem*, Malaysian *Tapai*, Korean *Makkulli* and *Yakju*, and Chinese *Huang jiu*, *Huadiao jiu*, *Shaohsing*, *Chia Fan*, *Hsiang Hsueh*, *Shan Niang*, and *Yen Hung* [17]. *Sake* is the traditional alcoholic rice beverage prepared and consumed in Japan and China. The seed mash for *Sake* preparation is traditionally obtained by natural lactic acid fermentation involving various aerobic bacteria, wild yeasts, LAB, and sake yeast [8]. During *Sake* brewing, saccharification and alcoholic fermentation progress simultaneously. The starter, *koji*, is rich in enzyme activity including amylases and proteases. From a nutritional point of view, *Sake* is distinctive among alcoholic beverages as it contains more proteins (0.4 g/100 g) and carbohydrates (4.1 g/100 g) that make it richer in taste than other beverages [14]. The rice wines *ruou de* or *ruou nep*, fermented from rice or glutinous rice, are popular traditional alcoholic beverages in Vietnam. The starters used for fermenting *ruou de* or *ruou nep* include yeasts, molds, and bacteria that convert starch material to fermentable sugars, which are subsequently converted to alcohol and organic acids. The alcohol content of these traditional undistilled wines is around 7%–10% (*v/v*) [15].

Herbs are known as rich sources of bioactive compounds entering in the preparation of traditional beverages (antioxidant, anti-inflammatory, antimicrobial). Traditional date juice (*Phoenix dactylifera*), Tassabount in Morocco, is a preparation using medicinal and aromatic plant macerate which is fermented for 3–5 days. A variety of plants are used including more than 20 species (basil, clove, thyme, lemon, iris, mythe, oregano, nutmeg, rosemary, mandrak). The fermentation process has the potential to produce new beneficial compounds, resulting in the increase of biological properties of traditional preparations. This aromatic extract has been reported to contain bioactive compounds such

as carvacrol, thymol, and phenolic compounds providing antimicrobial properties and improving the safety status and shelf life of the traditional juice [18,19].

*Angkak* or red yeast rice has been used extensively in Asian cuisine as a natural food colorant in fish, Chinese cheese, red wine, and sausages. *Angkak* is a product of the solid-state fermentation of rice by *Monasus* fungi which can convert starchy substrates into metabolites such as alcohols, antibiotic agents, antihypertensives, enzymes, fatty acids, flavor compounds, organic acids, pigments, and vitamins. *Angkak* contains mevinolin, a compound that inhibits cholesterol production by blocking a key enzyme, HMG-Coa reductase. However, another secondary metabolite, known as citrinin ($C_{13}H_{14}O_5$) which is a hepato-nephrotoxin, is also synthesized by *Monascus* strains [20]. The new product from an adlay substrate (Chinese pearl barley) fermented by *Monascus purpureus* produced the adlay angkak with the lowest citrinin and the highest mevinolin content [21]. The study showed the positive effect of the fermented papaya preparation (FPP) on Type 2 diabetes [22]. FPP was made from the yeast fermentation of ripen papaya using a specialized biotechnological technique. The evaluation was conducted by studying its effect on the human antioxidant status and erythrocyte integrity of a multi-ethnical pre-diabetic population. Fermented papaya exhibited effective in vitro free radical scavenging activities, believed to be attributed to the residual phenolic or flavonoid compounds.

### 2.2. Lactic Acid Fermentation

Lactic acid bacteria (LAB) belong to a group of Gram-positive facultative anaerobic bacteria that synthesize lactic acid as their main product of fermentation into the culture medium. Lactic acid fermentation is performed by LAB of which the genera are mostly composed of *Lactobacillus*, *Lactococcus*, *Enterococcus*, *Streptococcus*, *Pediococcus*, *Leuconostoc*, *Weisiella*, etc. Lactic acid bacteria assist in preserving and producing a wide range of foods. Traditional uses of many LAB as fermentation agents for foods are considered to be safe for the general population. Lactic acid bacteria cause rapid acidification of food due to the production of acids, primarily lactic acid. Other metabolites associated with LAB include acetic acid, ethanol, aromatic compounds, bacteriocins, exopolysaccharides, and several enzymes. These compounds result in the enhancement of shelf life and microbial safety, as well as the improvement of texture and sensory profile of the fermented products. Different fermented products are available that come mainly from lactic acid fermentation: fermented dairy products (cheese, butter, and butter milk, yoghurt, fermented probiotic milk, kefir); fermented meat products (sausages); fermented fish products; fermented fresh vegetables such as cabbage (sauerkraut, *kimchi*); cucumbers (pickles); fermented cereals (sourdough bread and bread-like products); and alcoholic beverages (wine) [23]. Moreover, because of their ability to produce lactic acid, up to 50 different species of *Lactobacillus plantarum* have been applied to popular traditional fermentation food technology for products such as meat, vegetables and dairy. Furthermore, improving the conversion, flavor and texture characteristics of fermented food are considered as main reasons for using this probiotic strain in industrial food technology. Tailor-made LABs with desired physiological traits can be constructed and applied to optimize the food manufacturing processes or to manipulate the organoleptic properties (i.e., the overall flavor and texture) of the products. Table 3 summarizes the common lactic acid-fermented food products in Asia.

*Kefir* is a traditional milk product, the combination of lactic acid bacteria and yeast. Special taste and aroma were found in a kefir product, the combination of lactic acid and alcohol. In addition, fresh milk was usually fermented by the kefir grain, insoluble in water and containing a group of microorganisms. Moreover, *Lactobacillus* species are considered as the majority of bacteria in kefir. Therefore, the kefir product was recognized as containing antimicrobial activity because kefir grain produces antibacterial metabolites such as lactic acid, volatile acids, hydrogen peroxide, carbon dioxide, diacetyl, and acetaldehyde. Many studies have focused on research into the survival capacity of LAB in cereal and fruit materials such as oat flour, malt, barley, and cabbage juice. The fruits and vegetables are considered as rich functional components such as minerals, vitamins, dietary fiber, and antioxidants. It is recognized that lactic acid milk fermentation can generate a large number of

peptides with potentially bioactive properties. Quirós et al. [26] fermented milk with different strains of *E. faecalis* and identified two peptides, corresponding to β-casein *f* (133–138) and β-casein *f* (58–76). These peptides demonstrated angiotensin converting enzyme-inhibitory (ACEI) activity. Furthermore, when administered orally to hypertensive rats, the peptides exhibited antihypertensive activity.

**Table 3.** Common lactic acid-fermented food products in Asia.

| Country | Raw Materials | Local Name | Microorganism | References |
|---|---|---|---|---|
| **Cereal Grains** | | | | |
| Nepal | Rice | *Selroti* | *Lactobacillus curvatus,*<br>*Pseudomonas pentosaceuts,*<br>*Escherecia faecium,*<br>*Saccharomyces cerevisiae,*<br>*Saccharomyces kluyveri,*<br>*Debaryomyces hansenii,*<br>*Pichia burtonii,*<br>*Zygosaccharomyces rouxii* | [24] |
| India | Rice<br>Black gram | *Idli*<br>*Dosa* | *Leuconostoc mesenteroides,*<br>*Streptococcus faecali,*<br>*Pediococcus cerevisiae* | |
| **Vegetables** | | | | |
| Nepal | Cabbage,<br>cauliflower, radish,<br>mustard leaf | *Gundruk* | *Pediococcus pentasaceous,*<br>*Lactobacillus cellubiosus,* and<br>*Lactobacillus plantarum* | |
| Korea | Cabbage, radish,<br>various vegetables | *Kimchi* | *Leuconostoc mesenteroides,*<br>*Lactobacillus brevis, Lactobacillus*<br>*plantarum* | |
| Vietnam | Cabbage | *Dhamuoi* | *Leuconostoc. mesenteroides,*<br>*Lactobacillus plantarum* | |
| Thailand | Mustard leaf | *Dakguadong* | *Lactobacillus plantarum* | |
| Philippines | Mustard leaf | *Burong*<br>*mustala* | *Lactobacillus brevis, Pediococcus*<br>*cerevisiae* | |
| **Meat and Fish Products** | | | | |
| Philippines | Fresh water fish,<br>rice, | *Burong-isda* | *Lactobacillus brevis*<br>*Streptococcus* sp. | |
| Thailand | Fresh water fish,<br>salt, steamed rice | *Som-fak*<br>*Pla-ra* | *Pediococcus* sp. and *Lactobacillus*<br>sp. | [25] |
| | Shrimp, salt,<br>sweetened rice | *Kungchao* | *Pediococcus cerevisiae* | |
| | Pork, garlic, salt,<br>rice | *Nham* | *Pediococcus cerevisiae*<br>*Lactobacillus plantarum*<br>*Lactobacillus brevis* | |
| Korea | Sea water fish,<br>cooked millet, salt | *Sikhae* | *Leuconostoc mesenteroides*<br>*Lactobacillus plantarum* | |
| Japan | Sea water fish,<br>cooked millet, salt | *Narezushi* | *Leuconostoc mesenteroides*<br>*Lactobacillus plantarum* | |
| Vietnam | Pork, salt, cooked<br>rice | *Nem-chua* | *Pediococcus* sp.<br>*Lactobacillus* sp. | |

Cereal products can be fermented either to produce alcoholic beverages or non-alcoholic food products mostly in the form of breads, loaves, confectionery, and gruels, or as complementary foods for infants and children. Non-alcoholic fermented cereal products include sourdough of America,

Europe, Australia; *dosa* and *idli* of India; *puto* of Southeast Asia; *masa* of South Africa; *kisra* of Sudan; *tarhana* of Turkey, and so on. Among the Nepalese people, fermented cereal food called *selroti*, which is prepared from rice, is quite popular. Yonzan and Tamang [24] studied the microbiology and nutritional values of *selroti* with the long-term goal of developing a good starter culture technology. The LAB strain, comprising lactobacilli, pediococci, leuconostocs, and enterococci, and the yeast strains *S. cerevisiae*, *S. kluyveri*, *D. hansenii*, *P. burtonii*, and *Z. rouxii* were found to co-exist as the predominant microorganisms. With fermentation, water-soluble and tricholoroacetic acid (TCA) soluble nitrogen were observed to increase, indicating an enhancement in protein digestibility.

Fermentation enhances the functional properties as well as the food value of the product. Indian *idli* and *dosa* are the traditional products made from polished rice and dehulled black gram by lactic acid fermentation. Both are products of natural lactic acid fermentation by *L. mesenteroides* and *S. faecalis*. Within the fermentation time of 20 h, the reduction of sugars (such as glucose) was observed, decreasing from 3.3 mg/g of dry ingredients to 0.8 mg/g, reflecting the utilization of sugar for acid and gas production. Flatulence-causing oligosaccharides, such as stachyose and raffinose, were found completely hydrolyzed. A decrease in phytate phosphorous and an increase in thiamine and riboflavin have also been reported during fermentation in the same study.

*Gundruk* is a non-salted fermented and dried vegetable product of Nepal. It is prepared by spontaneous lactic acid fermentation of green leafy vegetables, including the leaves of *Brassica* species such as mustard (*Brassica campestris* L.), rayo (*B. juncea* L.), cauliflower (*B. compestris* L. var. *botrytis* L.), cabbage (*B. oleracea* L, and radish (*Raphanus sativus* L.) [27]. *Gundruk* is one of the highly priced indigenous products of Nepal. It is served as a side dish with the main meal, as an appetizer, and as a soup. The annual production of *Gundruk* in Nepal is estimated at 2000 tons and most of the production is carried out at the household level. *Pediococcus pentasaceous*, *Lactobacillus cellubiosus*, and *Lactobacillus plantarum* are the dominant microorganisms in *Gundruk* fermentation. The pH values of *Gundruk* prepared from mustard, rape, and radish leaves were 4.0, 4.3, and 4.1, respectively, and the lactic acid contents were 1.0, 0.8, and 0.9%, respectively [12].

Traditionally fermented lightly salted fish products composed of fish, salt (2%–7%), carbohydrate source (rice, millet, sugar), and spices (garlic, ginger, chili, pepper) are popular in Southeast Asia. *Som-fak* is a traditional Thai product composed of fish fillet, salt (2%–5%), ground boiled rice (2%–12%), and minced garlic (4%). The mixtures are tightly packed in banana leaves and allowed to ferment at ambient temperature for 2–4 days. Rapid growth of LAB drops pH below 4.5, and *Pediococcus sp.* and *Lactobacillus* sp. have been identified as the dominating LAB [28]. Ribas-Agustí et al. [29] developed fermented sausage with the addition of vegetable extract, cocoa extract, and grape seed extract (GSE) with the aim of producing fermented meat with a balanced quantity of phenolic and other bio-active compounds. After completing the aging process, catechin and epicatechin were at 54%–61%, gallic acid and galloylated flavan-3-ols were at 59%–91%, oligomeric flavan-3-ols were at 72%–95%, and glycosylated flavonols were at 56%–88% (in cocoa treatment), as well as 82%–94% (in GSE treatment) of the contents that were added to the meat batter.

## 2.3. Acetic Acid Fermentation

Acetic acid bacteria (AAB) are also commonly found in a wide range of fermented foods and beverages. AAB are ubiquitous, aerobic, Gram-negative bacteria belonging to the *Acetobacteraceae*, and the genera *Acetobacter*, *Gluconobacter*, *Gluconacetobacter*, and *Komagataeibacter* constitute the common AAB found in food and beverage fermentation such as cocoa, milk kefir, water kefir, kombucha, and acidic beers [30]. AAB are predominantly known for their use in the production of vinegar, vitamin C, and cellulose. The food substrate rich in carbohydrates, sugar, alcohols, and/or ethanol enables AAB to rapidly and incompletely oxidize these substrates into organic acids (acetic acid). However, AAB are not studied to the same extent as many other food-grade and industrially important microorganisms. Moreover, AAB are regarded as undesirable spoilers in alcoholic fermentation [31].

Kombucha from Central and East Asia is a beverage obtained by the fermentation of sweetened boiled tea with a mixed culture of yeasts and acetic acid bacteria. Other names for kombucha, or "tea fungus", include "fungus japonicus", "tee kwass", "tea kvass", "champignon de longue vie", "Indo-Japanese tea fungus", and "Manchurian mushroom" [32]. The microbiological composition of the tea fungus exhibits the symbiosis between bacteria and fungus. The main acetic acid bacteria found in the tea fungus are the following: *Acetobacter xylinum*, *A. xylinoides*, *A. acetic*, *A. pasteuriansu*, *Bacterium gluconicum*. Yeasts isolated from tea fungus include *Schizosaccharomyce pombe*, *Saccharomycodes ludwiggi*, *Kloeckera apiculate*, *Saccharomyces cerevisiae*, *Zygosaccharomyces bailii*, *Brettanomyces bruxellensis*, *B. lambicus*, *B. custersii*, *Candida* and *Pichia* species. The yeast cells convert sucrose into fructose and glucose and produce ethanol. Acetic acid bacteria (AAB) convert glucose to gluconic acid and fructose into acetic acid. Caffeine and related xanthines of the tea infusion stimulate the cellulose synthesis by the bacteria. Acetic acid stimulates the yeast to produce ethanol and ethanol which in turn stimulate AAB to produce acetic acid [33].

## 2.4. Alkaline Fermentation

Alkaline-fermented food products play an important role in the diet of people from Asia, Africa, and worldwide. Protein-rich foods are the main substrate that are acted upon primarily by *Bacillus* spp., but other secondary microorganisms such as LAB, staphylococci, and micrococci are also involved. Alkaline fermentation involves proteolysis that releases peptides and essential amino acids. Furthermore, amino acids are degraded into alkaline compounds such as ammonia that cause an increase in pH (8–10). Some of the common alkaline-fermented foods include *soumbala*, *ugha*, *bikalga*, and *ntoba mbodi* from Africa; as well as *kinema*, *natto*, and *thua nao* from Asia [34]. Species of *Bacillus* that are present, mostly in legume-based fermented foods, are *Bacillus amyloliquefaciens*, *B. circulans*, *B. coagulans*, *B. firmus*, *B. licheniformis*, *B. megaterium*, *B. pumilus*, *B. subtilis*, *B. subtilis* var. *natto*, and *B. thuringiensis*, whereas strains of *B. cereus* have been isolated from the fermentation of *Prosopis africana* seeds for the production of *okpehe* in Nigeria. Some strains of *B. subtilis* produce λ-polyglutamic acid (PGA), which is an amino acid polymer commonly present in Asian fermented soybean foods, giving the characteristic of a sticky texture to the product.

Soybean is one of the common substrates for traditional alkaline fermentation in East and Southeast Asia and in West Africa. The main microorganism for alkaline fermentation of non-salted soybean involves *Bacillus subtilis*. During fermentation, protease and amylase enzymes act upon protein and insoluble sugar, and therefore improve the nutritional value of fermented soybean products. Compared to the non-fermented counterpart, fermented soybeans are rich in isoflavone genestein and gamma-polyglutamic acid (PGA). Isoflavone genestein acts as a chemopreventive agent against cancer, while PGA acts as dietary fiber to reduce the cholesterol level in serum and improves efficacy of calcium absorption [35]. Soybean products can be either fermented by *Bacillus* spp. (mostly *Bacillus subtilis*) or by filamentous molds (mostly *Aspergillus*, *Mucor*, *Rhizopus*). Non-salted and sticky soybean products fermented by *Bacillus* spp. are concentrated in an imaginary triangle with three vertices lying each on Japan (*natto*), eastern Nepal and north-eastern India (*kinema*), and northern Thailand (*thua nao*), named as the "natto triangle" or renamed as the "kinema-natto-thuanao (KNT)-triangle". Traditional *Bacillus* fermented soybean products other than this include *chungkokjang* of Korea, *aakhune*, *bekang*, *hawaijar*, *peruyaan*, and *tungrymbai* of India, *pepok* of Myanmar, and *sieng* of Cambodia and Laos [10].

*Natto* is one of the most popular alkaline-fermented soybean products consumed widely in Japan, such that one Japanese consumes 760 g of natto per year. It is served along with rice as breakfast or as flavoring in dishes. The distinct features of *natto* include its unique odor, flavor, and notably stringy, mucous material on the surface of the soybean. The *Natto* preparation method includes cleaning, soaking, and cooking of the soybean followed by inoculating it with *B. subtilis* var. *natto* and fermentation at 40 to 45 °C for 18 to 20 h. The stringy material formed during fermentation is a polypeptide of glutamic acid and fructan produced by *B. subtilis* var. *natto*. *Natto* contains nattokinase, a polypeptide composed of a total of 27 amino acid residues with anticoagulant, fibrinolytic, and blood pressure-lowering effects, and antioxidant activity [36].

*Thua nao*, an alkaline-fermented soybean product, is a rich source of free amino acids and is used as a condiment in northern Thailand. Traditionally, it is prepared by fermenting the boiled soybean (boiled for 3–4 h) in bamboo baskets covered with banana leaves. The fermentation at ambient temperature for three days results in a brownish slimy substance with an ammonia odor and which is rich in free amino acids [37]. The dominant microflora of *thua nao* was found to be *Bacillus* spp., a Gram-positive, strict or facultative aerobe and endospore-forming bacteria [38]. Similar, *kinema* is an alkaline soybean fermented product that constitutes a large part of the diet of people living in eastern Nepal, and Darjeeling and Sikkim in India. Traditionally, *kinema* is prepared from clean soybeans soaked overnight, followed by cooking, crushing, wrapping in leaves and sackcloth, and fermenting at 25–35 °C for 1–3 days. It is an inexpensive source of nutrition containing (per kg; dry weight basis) 356–487 g protein, 161–249 g crude lipid, 274–296 g carbohydrate, and a number of vitamin B and minerals [39]. *Kinema* with a pungent ammonia smell, a slimy texture, and a short shelf-life, is similar to "*natto*" and "*thua nao*" with *Bacillus subtilis* as the dominant microorganisms. Compared to unfermented soybean, fermented *kinema* is found to be richer in essential fatty acids and better in protein quality and digestibility [40]. Another traditionally fermented soybean product that acts as a low-cost source of high protein food for the local people of Manipur, India is *Hawaijar*. It is similar to "natto" but with a dark brown color and a unique odor and taste. In research carried out by Jeyaram et al. [2], a *Bacillus subtilis* group comprising *B. subtilis* and *B. licheniformis*, and *B. cereus* and a *Staphylococcus* spp. group comprising *S. aureus* and *S. sciuri* were isolated from 41 "Hawaijar" samples collected from household preparations and markets of Manipur.

*Tempe*, a traditional soybean fermented product obtained through fermentation with *Rhizopus* spp., originates in Indonesia. During fermentation, different fungal enzymes including proteases, lipases, carbohydrases, and phyatses are produced that break down macromolecules into simpler compounds, thus partly solubilizing the cell walls and intracellular material. The final fermented product therefore has an increased nutritional quality and digestibility and the cottony mycelium binds the soybeans forming a compact cake. *Tempe* fermented by *Rhizopus microsporus* caused a reduction in the severity of diarrhea in piglets by inhibiting the adhesion of enterotoxigenic *Escherichia coli* to intestinal brush border cells [41]. Table 4 summarizes the common alkaline-fermented foods and beverages of Asia.

**Table 4.** Some of the common alkaline-fermented foods and beverages in Asia.

| Country | Raw Materials | Local Name | Microorganism | References |
|---------|---------------|------------|---------------|------------|
| Thailand | Soybean | *Thua nao* | *Bacillus subtilis* | [35] |
| India | Soybean | *Hawaijar* | *Bacillus* spp. | [2] |
| Japan | Soybean | *Natto* | *Bacillus subtilis* var. *natto* | [36] |
| Korea | Soybean | *Cheonggukjang* | *Bacillus Subtilis* | |
| Nepal | Soybean | *Kinema* | *Bacillus subtilis* | [40] |

## 3. Conclusions and Outlook

In conclusion, Asia is well-known for its exotic traditionally fermented food and beverage products produced using a wide range of raw materials, microorganisms and fermentation processes. The indigenous methods of fermentation were aimed to preserve and balance the availability of food sources. Furthermore, many scientific research studies have exhibited promising and sustainable opportunities related to these traditionally fermented food products. The nutritional values of fermented foods are related to a unique group of microflora that may enhance health benefits directly through the interaction with the host or indirectly through metabolites synthesized during fermentation. The bioactive compounds and other interactions within fermented food can add novel flavors to food and impart potential health benefits. However, future studies need to be conducted in order to explore various aspects of fermented food products, such as the determination of biomarkers for fermented food health benefits, safety concerns related to these products, and the bioaccessiblity of microbial metabolites.

**Funding:** This research received no external funding.

**Conflicts of Interest:** The author declares no conflict of interest.

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
