# Peer review of "Quality Ingredients and Safety Concerns for Traditional Fermented Foods and Beverages from Asia: A Review"

_fermentation, doi:10.3390/fermentation5010008_

Round 1
Reviewer 1 Report
Fig 1, legend and text related to are bit unclear. For example, the alcoholic fermentation produce a fermented food (alcoholic beverages) while the lactic acid fermentation produce a metabolite (lactic acid). Moreover, not all the alcoholic fermentation with yeast produce alcoholic beverages.
-Line 211. LAB has been already abbreviate
-Line 225. ....50 strains of L. plantarum, I guess. Probably some references may be added.
-Line 228-229. Unclear..."this probiotic strain??"
- Table 3. Local names of fermented product, I guess. Please check species names. Some of them are wrong, other are incorrectly written
- Line 253-257. Please editing
- Conclusions and critical suggestion on traditional food fermentation may be reported/added (e.g., are all these fermentation process safe? Do they use microbial starter? Do we need to select microbial starter?...)
Author Response
Reviewer 1:
Comment 1: Fig 1, legend and text related to are bit unclear. For example, the alcoholic fermentation produce a fermented food (alcoholic beverages) while the lactic acid fermentation produce a metabolite (lactic acid). Moreover, not all the alcoholic fermentation with yeast produce alcoholic beverages.
Response: Thank you; It has been changed and made it clear in manuscript as suggested.
Comment 2: Line 211. LAB has been already abbreviated
Response: Though LAB has already been abbreviated in previous section, It is still written full form again as this sentence starts with Lactobacillus.
Comment3: -Line 225. ....50 strains of L. plantarum, I guess. Probably some references may be added.
Response: Thank you . This should be 50 different species of LAB and it is modified accordingly in manuscript.
Comment 4: Line 228-229. Unclear..."this probiotic strain??"
Response: This line has been deleted from the manuscript. Thanks for suggestion
Comment 5: Table 3. Local names of fermented product, I guess. Please check species names. Some of them are wrong, other are incorrectly writte
Response: The local names of fermented products are rechecked and written with corrects names.
Comment 6: Line 253-257. Please editing
Response: It has now been rewritten to make very clear.
Comment7: Conclusions and critical suggestion on traditional food fermentation may be reported/added (e.g., are all these fermentation process safe? Do they use microbial starter? Do we need to select microbial starter?...)
Response: Thank you very much for this suggestion,. The Conclusion and Future Outlook Section is included at the end of the manuscript.
Reviewer 2 Report
Line 26: "fermentation technique" sounds a bit weird. Maybe you could just say fermentation.
Line 39: basic fermentation processes-this statement is repeated in the next line/paragraph.
Table 1: this table is not quite understandable... "Fermentation of the final food product" - for example beer of wine... they ARE the final products of alcoholic fermentation. Please explain.
Figure 1: repeats everything you stated at the beginning of the introduction. I find it unnecessary.
Line 89: "The main four types of fermentation processes include: alcoholic, lactic acid, acetic acid and alkali fermentation." This was stated several time, and there is no need to repeat this.
Lines 91-92: "During alcoholic fermentation, yeasts are the predominant microorganisms and results in the production of ethanol." This sentence feels off here. Maybe transfer it to the next paragraph.
Line 147: % should be next to the number.
Table 3: please align the columns.
References: all of the references are missing doi.
There are too little references in the text. Big paragraphs with only one reference... you should more precisely specify the part from other literature. For example: the classification in the introduction (Lines 41-47)... there are many such parts, so please be more specific in citing literature.
The English language should be checked by and English speaker or professional. It is generally OK, but needs to be improved (mainly missing articles, etc.).
Author Response
Reviewer 2
Comment1: Line 26: "fermentation technique" sounds a bit weird. Maybe you could just say fermentation.
Response: Thank you. Agreed and word has now been changed to Fermentation only.
Comment 2: Line 39: basic fermentation processes-this statement is repeated in the next line/paragraph.
Response: Thank you. It has been taken consideration and the repeated words are deleted.
Comment 3: Table 1: this table is not quite understandable... "Fermentation of the final food product" - for example beer of wine... they ARE the final products of alcoholic fermentation. Please explain.
Response: Thank you, The titel of the table is changed now accordingly for more clear.
Comment 4: Figure 1: repeats everything you stated at the beginning of the introduction. I find it unnecessary.
Response: As Figure 1 illustartes briefly the proceses and end products of fermentation, I prefer to keep this as gives clear pictures to reader immediately.
Comment 5: Line 89: "The main four types of fermentation processes include: alcoholic, lactic acid, acetic acid and alkali fermentation." This was stated several time, and there is no need to repeat this.
Response: Thank you. This has been taken consideration and repeated words/sentences have been deleted in manuscript.
Comment 6: Lines 91-92: "During alcoholic fermentation, yeasts are the predominant microorganisms and results in the production of ethanol." This sentence feels off here. Maybe transfer it to the next paragraph.
Response: Thank you. As suggested, it has been moved to next paragraph.
Comment 7: Line 147: % should be next to the number.
Response: thank you, it has been corrected.
Comment 8: Table 3: please align the columns.
Response: Thank you. The table has been aligned as suggested.
Comment 9: References: all of the references are missing doi.
Response: As all these references are having issue, volume and page numbers, it is not necessary to provide DOI.
Comment 10: There are too little references in the text. Big paragraphs with only one reference... you should more precisely specify the part from other literature. For example: the classification in the introduction (Lines 41-47)... there are many such parts, so please be more specific in citing literature.
Reference: Thank you, we have included some references in between as these references are already listed either in tables and/or in texts.
Comment 11: The English language should be checked by and English speaker or professional. It is generally OK, but needs to be improved (mainly missing articles, etc.).
Response: The English has been checked thoroughly with the help of Native English speaker and corrected accordingly.
Round 2
Reviewer 1 Report
paper has been improved